# Survival rate, blood feeding habits and sibling species composition of *Aedes simpsoni complex*: Implications for arbovirus transmission risk in East Africa

Winnie W. Kamau[1,2], Rosemary Sang[1,3], Edwin O. Ogola[1], Gilbert Rotich[1], Caroline Getugi[1], Sheila B. Agha[1], Nelson Menza[2], Baldwyn Torto[1], David P. Tchouassi[1] *

1 International Centre of Insect Physiology and Ecology, Nairobi, Kenya, 2 Kenyatta University, Nairobi, Kenya, 3 Center for Virus Research, Kenya Medical Research Institute, Nairobi, Kenya

* dtchouassi@icipe.org

**Data Availability Statement:** Sequences generated in this work have been deposited in the GenBank under accession numbers OK339453 – OK339472 and OK339480 – OK339521 (*Aedes bromeliae*);

## Abstract

*Aedes simpsoni* complex has a wide distribution in Africa and comprises at least three described sub-species including the yellow fever virus (YFV) vector *Ae. bromeliae*. To date, the distribution and relative contributions of the sub-species and/or subpopulations including bionomic characteristics in relation to YF transmission dynamics remain poorly studied. In this study conducted in two areas with divergent ecosystems: peri-urban (coastal Rabai) and rural (Rift Valley Kerio Valley) in Kenya, survival rate was estimated by parity in *Ae. simpsoni s.l.* mosquitoes sampled using $CO_2$-baited BG Sentinel traps. We then applied PCR targeting the nuclear internal transcribed spacer 2 (ITS2), region followed by sequencing and phylogenetic analytics to identify the sibling species in the *Ae. simpsoni* complex among parous and blood fed cohorts. Our results show that *Ae. bromeliae* was the most dominant sub-species in both areas, exhibiting high survival rates, human blood-feeding, and potentially, high vectorial capacity for pathogen transmission. We document for the first time the presence of *Ae. lilii* in Kenya and potentially yet-to-be described species in the complex displaying human feeding tendencies. We also infer a wide host feeding range on rodents, reptile, and domestic livestock besides humans especially for *Ae. bromeliae*. This feeding trend could likely expose humans to various zoonotic pathogens. Taken together, we highlight the utility of genotype-based analyses to generate precision surveillance data of vector populations for enhanced disease risk prediction and to guide cost-effective interventions (e.g. YF vaccinations).

## Author summary

Yellow fever (YF) remains a significant public health risk in East Africa, however, with gaps in the transmission ecology. Important YF virus vectors include *Aedes simpsoni* mosquitoes that comprise subspecies with varying vectoring abilities and, poorly described

OK339522 – OK339529 (*Aedes lilii*); OK339473 – OK339479 (*Aedes simpsoni s.l.*).

**Funding:** WWK received an ICIPE MSc-Dissertation Research Internship Programme (DRIP) scholarship through the project Combatting Arthropod Pests for better Health, Food and Climate Resilience (CAP-Africa; project number RAF-3058 KEN-18/0005) funded by Norwegian Agency for Development Cooperation (Norad). Financial support for the study came from the Norad funded CAP-Africa project. DPT is also supported by a Wellcome Trust International Intermediate Fellowship (grant number 222005/Z/20/Z). We also gratefully acknowledge the financial support and technical support of ICIPE core donors: Swiss Agency for Development and Cooperation (SDC), Switzerland; Swedish International Development Cooperation Agency (Sida), Sweden; Ministry of Higher Education, Science and Technology, Kenya; and Government of the Federal Democratic Republic of Ethiopia. The views expressed herein do not necessarily reflect the official opinion of the donors. The funders had no role in study design, data collection and analysis, decision to publish, or preparation of the manuscript.

**Competing interests:** The authors have declared that no competing interests exist.

ecology and biologic traits relevant to disease transmission. Through active surveillance, we analyzed the survival, human blood feeding habits and genetics of wild populations of *Aedes simpsoni s.l.* in two contrasting ecosystems in Kenya: peri-urban, coastal Rabai, and rural, Rift Valley Kerio Valley. Our findings reveal i) *Aedes bromeliae* as the most abundant subspecies in both areas exhibiting high survival rates, human blood-feeding, and potentially, high vectoring ability, ii) occurrence of *Ae. lilii* contrary to previous reports albeit in low numbers, iii) potential undescribed species in the group displaying human feeding tendencies. Knowledge of the locally adapted subspecies and associated traits that underlie vectorial capacity, impinges on YF distribution risk useful for guiding vector control or cost-effective vaccination.

## Introduction

Yellow fever (YF) is a re-emerging arboviral threat in Africa and South America despite availability of an effective vaccine to protect humans. This is unparalleled in eastern Africa exemplified by increased frequency and magnitude of outbreaks recorded recently in Sudan (2012), South Sudan (2020), Uganda (2011, 2016, 2019, 2020), and Ethiopia (2012–2014, 2018; 2020) [1,2]. Imported cases into Kenya following recent outbreaks in Angola and the Democratic Republic of Congo (DRC) (2015–2016) [2] demonstrate the potential for continued YF spread in the region. Improved understanding of the drivers of YF virus (YFV) transmission are urgently required through risk assessment to inform cost-effective preventive strategies.

Of the recognized YFV transmission cycles (sylvatic, rural, and urban cycles), outbreaks in the eastern African region have been described as sylvatic [3]. This is inclusive of the last documented YF outbreak in Kenya (1992–95) that implicated sylvatic *Aedes* vectors including *Aedes simpsoni s.l.* [4]. However, YFV transmission trends appear to be changing with demographic and environmental changes. For instance, the 2015/16 outbreaks in Angola and DRC were urban [2] and whether the East Africa region may face same fate in the future remains uncertain. The abundance of *Ae. simpsoni s.l.* vectors in peri-domestic human environments and associated Stegomyia indices have raised concerns about potential urban/rural risk of YFV transmission in Kenya [5]. *Aedes simpsoni s.l.* thus, has the potential to serve as bridge vector, moving the YFV from the sylvatic/rural to the urban transmission cycle. *Aedes simpsoni* complex comprises at least three described species including; *Ae. lilii, Ae. simpsoni s.s.*, and the YF vector *Ae. bromeliae* [6]. To date, the distribution and relative contributions of the sub-species and/or subpopulations including bionomic characteristics in relation to YFV transmission dynamics remain poorly studied.

Demographic and societal changes including unplanned urbanization as occurring in Africa [7], are among important drivers of vector-borne disease spread. The consequent changes in urban architecture, via effects on environment, human abundance, vector density, and biting behavior have the potential to alter the vectorial capacity of mosquitoes and diseases transmission risk. We here, report data on survival and host feeding patterns of *Ae. simpsoni* mosquitoes in two areas of contrasting ecosystems: peri-urban, coastal Rabai, and rural, Rift Valley Kerio Valley, in Kenya. These parameters provide opportunities for vector infection and efficient transmission and are the two most sensitive indicators of disease transmission potential (i.e. vectorial capacity) [8,9]. Because, human vector feeding must precede outbreaks, knowledge of vector host-feeding behavior is essential to our understanding of the inter-epidemic maintenance patterns of YF. Furthermore, we assessed how these bionomic attributes vary among subspecies of the *Ae. simpsoni* complex.

## Materials and methods

### Ethics statement

The study received approval from the Scientific Ethics Review Unit (SERU) of the Kenya Medical Research Institute (Protocol NO. SSC 2787). Additionally, consent was sought verbally from household heads to set up traps around their homesteads.

### Study site

Adult female *Aedes simpsoni* senso lato (*s.l.*) (hereafter as *Ae. simpsoni*) mosquitoes that had been collected from Rabai (peri-urban environment; human density ~600/km$^2$) and from rural/sylvatic Kerio Valley (KV) (human density ~ 45/km$^2$) as part of an arbovirus surveillance project were used in this study. Rabai (Kilifi County) is located northeast and ~25 km from Mombasa City in coastal Kenya while Kerio Valley with a history of YF outbreak [4] in the Rift Valley (Fig 1). The mosquito breeding habitats differ with Kerio Valley providing rural woodland setting with numerous tree holes for mosquito breeding as opposed to Rabai, with the peri-urban setting providing typical water storage containers for mosquitoes to breed and also outdoors breeding in plant axils and water receptables. Kerio Valley is an arid to semi-arid ecology that is sparsely populated. The main economic activities of the inhabitants include farming of crops like maize and cotton, and livestock keeping especially goats and cattle. Rabai is one of the seven administrative sub-counties of Kilifi county. The main economic activities in the area include subsistence agriculture, casual labor, crafts and petty trading.

### Mosquito survey

Adult host seeking female mosquitoes were surveyed using $CO_2$-baited BG Sentinel traps at three trapping periods with trapping exercise beginning about three weeks after the area first received rainfall continuously for at least a week. This timing was critical to maximize assessment of vector abundance, considering the predominant tree hole and plant axils breeding ecology of *Ae. simpsoni* [10]. The trapping periods were August-September 2019 (mean daily temperature = 25.4˚C; mean daily rainfall = 0.58mm; relative humidity = 75.9%) and February 2020 (mean daily temperature = 27.4˚C; mean daily rainfall = 0.30mm; relative humidity = 80.3%) in Rabai and November 2019 (mean daily temperature = 19.9˚C; mean daily rainfall = 3.76mm; relative humidity = 72.4%) in KV. This trap type suitably targets *Stegomyia* mosquitoes which are active during the day [11]. Traps were set from morning to evening (06:30–18:00) on the same day, for at least 8 consecutive days per trapping period in peridomestic areas around homesteads. Sampling was designed to cover a large spatial area within each site. Traps were placed outdoors in the vegetation around human habitations and moved every other day to a new locality at least 500 m away. After retrieval, the mosquitoes were immobilised using triethylamine, sorted and transported in liquid nitrogen from the field to the Emerging Infectious Diseases laboratory at ICIPE, Nairobi. The mosquitoes were later identified morphologically to species level using published taxonomic keys [12] before storing in -80˚C freezers for later use.

### Parity rate, daily survival and longevity estimation

Mosquitoes after retrieval from -80˚C were allowed to thaw on ice (4˚C) and then dissected for parity after observing the degree of dilation of the tracheolar skeins of the avarioles [13]. After dissection, the remaining portion of each mosquito was preserved and processed for DNA extraction and molecular speciation (described below). Daily survival rates were derived from estimated parity rates for each sampling period as described previously [14] based on the

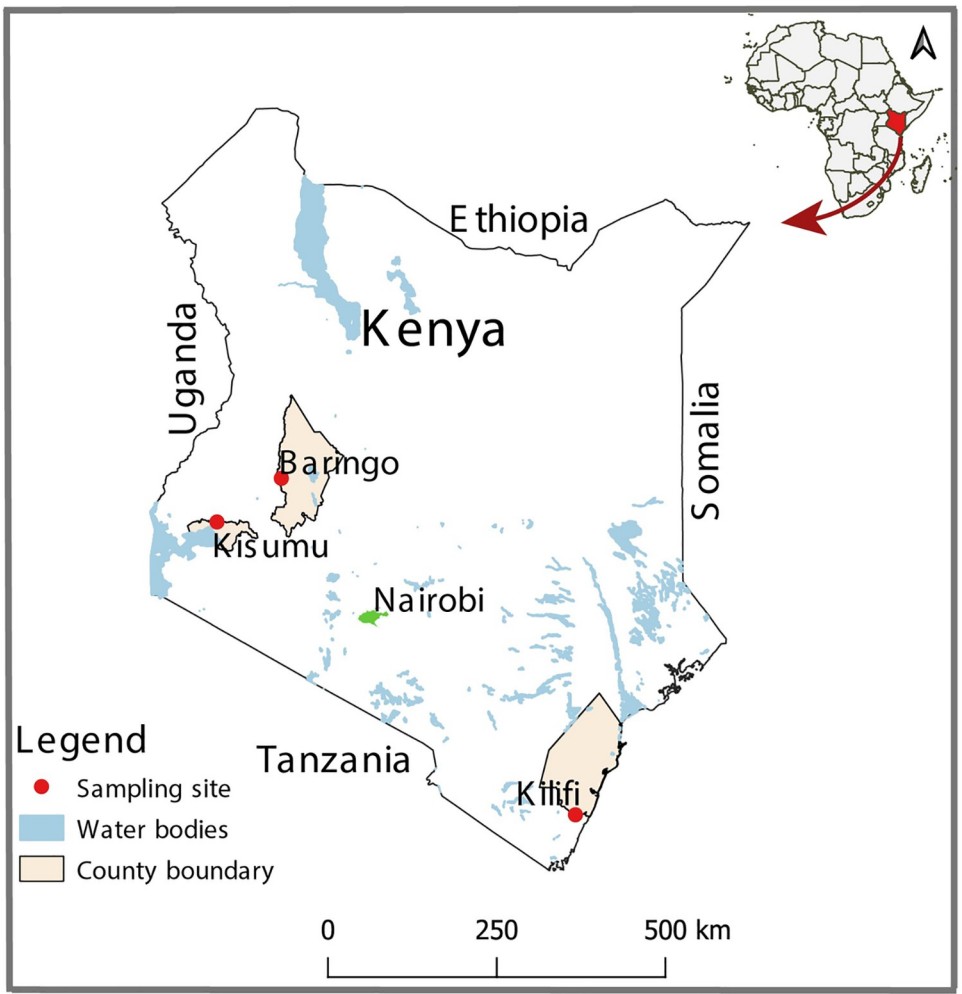

**Fig 1. Map of Kenya showing mosquito sampling sites: Kerio Valley (Baringo), Rabai (Kilifi) and Kajulu (Kisumu).** The map was designed using ArcMap 10.2.2 with the ocean and lakes base layer derived from Natural Earth (http://www.naturalearthdata.com/, a free GIS data source). The sample points were collected using a GPS gadget (garmin etrex 20, https://buy.garmin.com/en-US/US/p/518046), and the county boundaries for Kenya derived from Africa Open data (https://africaopendata.org/dataset/kenya-counties-shapefile, license Creative Commons).

formula: $P^n = M$, where P is the daily survival rate, $M$ the parity rate and $n$ the gonotrophic cycle (the number of days between emergence of adult females and first oviposition). A value of 3 days was assumed for this species [9]. Finally, the longevity (days) was estimated using the formula: $1/-ln$L where L is the estimated survival [15].

## Analysis of blood meals

Blood-fed specimens were individually dissected by separating the abdomen containing engorged blood from the head/thorax. The head/thorax were preserved for each mosquito and subjected to DNA extraction and molecular speciation (described below). Genomic DNA was extracted from the abdomen using the ISOLATE II Genomic DNA Kit (Bioline, Meridian Bioscience, Germany) as per the manufacturer's instructions. DNA was amplified targeting a 500 bp fragment of the 12S mitochondrial rRNA gene using the primers 12S3F [5'-GGGATTAGA-TACCCCACTATGC-3'] and 12S5R [5'-TGCTTACCATGTTACGACTT-3'] [16] and as

described previously [9]. PCRs in a 10 μl reaction volume comprised 2 μl 2x MyTaq Mix (Bioline, Meridian Bioscience, Germany), 10 μM of each primer, 0.2 U of Mytaq DNA polymerase and 1 μl of the template DNA (~20 ng). Thermal cycling conditions were 95˚C for 3 min followed by 40 cycles at 95˚C for 20 s, 59˚C for 30 s and 72˚C for 30 s and 72˚C for 7 min. Amplicons were resolved on 1.2% agarose gel electrophoresis against a 100bp DNA HyperLadder (Bioline, Meridian Bioscience, Tennessee, USA). The PCR products were purified using the SureClean Plus kit (Bioline, Meridian Bioscience) and outsourced to Microgen (Netherlands) for Sanger sequencing using the forward primer. DNA sequences were compared using the BLAST algorithm and the GenBank database (http://blast.ncbi.nlm.nih.gov/Blast.cgi). Species level identification was determined when sequences exhibited ≥ 98% identity spanning at least 300 bp as described previously [9].

## Species typing of mosquitoes

Genomic DNA was extracted from the head/thorax of each blood fed specimen and remaining portion of selected parous mosquitoes as described previously. PCR was performed to identify the sibling species of *Ae. simpsoni* targeting the nuclear internal transcribed spacer 2 (ITS2) region using the primers ITS2A (5'-TGTGAACTGCAGGACACAT-3') and ITS2B (5'-TATGCTTAAATTCAGGGGGT-3') [17]. ITS2 is fast evolving as a marker of choice widely used for species-level discrimination [18–20] with vast sequence representation in public databases (e.g. GenBank). PCRs were performed in 20μl reaction volumes including 3μl 5x HOT FIREPol Blend Master Mix Kit (Solis Biodyne, Tartu, Estonia), 0.5μl of 10 μM forward and reverse primers, of 5x Mytaq HS mix polymerase and 20 ng DNA template in a ProFlex PCR systems thermocycler (Applied Biosystems, Foster City, CA, USA). The thermal cycling conditions were 95˚C for 15 min followed by 40 cycles at 95˚C for 30 s, 60˚C for 30 s and 72˚C for 45 s and 72˚C for 7 min. Amplicons were confirmed by gel electrophoresis as described previously. Similarly, the PCR products were purified using SureClean Plus kit and outsourced for Sanger sequencing, in both the forward and reverse direction.

Sequences were viewed and edited in Chromas, embedded in MEGA v.6 [21] prior to querying the GenBank using BLASTn (www.ncbi.nlm.nih.gov/blast). Multiple sequence alignments of the resulting contiguous sequences were performed using ClustalW in MEGA v.6 with default parameters. Maximum likelihood (ML) trees were constructed with nodal support for the different groupings evaluated through 1000 bootstrap replications utilizing the Jukes and Cantor as best-fit model of sequence evolution.

## Statistical analyses

The parity rate calculated as the percentage of parous mosquitoes to the total number dissected was established for each trapping period and comparisons made by Pearson chi-squared tests. The 95% confidence intervals (CIs) for the parity rates were estimated using *binom.confint* function. The human blood index is expressed as the proportion of blood-feeding on humans of the total number of blood-fed mosquitoes examined and established for subspecies and tested for significant differences using Pearson chi-squared tests. All analyses were performed at P = 0.05 using R v. 4.0.4 software [22].

# Results

## Parity, daily survival and age estimates

Overall parity rate was 85% (458/539; 95% CI 81.7–87.7%) broken down as follows: Rabai: August-September 2019 = 82.6% (71/86; 95% CI 73.2–89.1%); February 2020 = 85.1% (189/

**Table 1. Estimated parity, daily survival and age for *Ae. simpsoni* mosquitoes.**

| Site | Survey period | Parity (% (n)) | Daily survival rate | Longevity (days) |
|---|---|---|---|---|
| Rabai | August-September 2019 | 82.6 (86)[a] | 0.9387 | 15.8 |
| | February 2020 | 85.1 (222)[a] | 0.9483 | 18.8 |
| Kerio Valley | November 2019 | 85.7 (231)[a] | 0.9504 | 19.7 |

a, not significant

222; 95% CI 79.9–89.2%); KV: November 2019 = 85.7% (198/231; 95% CI 80.6–89.6%). The parity rate did not differ between the sites or sampling periods (P>0.05; Table 1). Estimated daily survival rates for *Ae. simpsoni* were high across the sampling periods in Rabai translating to longevity ranging from 15.8–18.8 days (Table 1). *Aedes simpsoni* in Kerio Valley had a higher longevity compared to Rabai, though the difference was not significant (P>0.05; Table 1).

## Genetic composition of parous *Ae. simpsoni s.l.*

A subset of parous specimens (n = 43; Rabai = 37; KV = 6) randomly selected were profiled for DNA sequencing and phylogenetic analysis of the ITS2 region. Our findings show that most of the samples clustered with reference sequences of *Ae. bromeliae* with a strong bootstrap support (73%) (Fig 2). One specimen from Rabai grouped together with *Ae. lilii* in the Genbank and *Ae. simpsoni* samples included from Kajulu, in Kisumu County, western Kenya (Fig 2).

## Blood meal feeding patterns

A total of 27 blood fed *Ae. simpsoni s.l.* mosquitoes were analysed from Kerio Valley (n = 9) and Rabai (n = 18), of which blood meal sources were successfully identified from 24 specimens. These represented 9 specimens from Kerio Valley and 15 from Rabai. The data revealed a total of 10 different hosts belonging to fairly large mammals (humans, goat, cow, domestic cat), rodents (grass rat, squirrel, mastomys mouse, African giant pouched mouse, mongoose) and reptile (lizard) (Fig 3). The host range was more diverse in Rabai than Kerio Valley (Fig 3). In both areas, *Ae. simpsoni* had fed more on humans, then squirrel (Rabai) followed by a lower representation of other host types (Fig 3). The overall human-blood-index (HBI) was 0.33 (8/24).

## Association of *Ae. simpsoni* subspecies and human blood feeding

Next, we examined the relationship between subspecies of *Ae. simpsoni s.l.* and their influence on human blood feeding. A total of 27 *Ae. simpsoni sl* blood-fed specimens were profiled for phylogenetic analysis of sequenced ITS-2 region, with most of samples obtained from Rabai (n = 18), then Kerio Valley (n = 9). ITS2 sequences of the three blood-fed samples with unsuccessful blood meal data were included. The *Ae. simpsoni sl* samples resolved into 3 clades, with well supported bootstrap values (Fig 4). One of these clustered with *Aedes bromeliae* (GenBank No: KF135509) and had most of the samples from Rabai (n = 11) and Kerio Valley (n = 6). Clade II contained samples exclusively from Rabai (n = 4), while clade III had samples solely from Kerio Valley (n = 3). Thus, our findings show overwhelming representation of *Ae. bromeliae* among the blood-fed samples. Each of the clades exhibited human feeding tendency with no variation in the estimated human blood index (HBI) between the clades (P>0.05); clade I which clustered with *Ae. bromeliae*: (4/18); clade II: 2/3; clade III: 2/3 (Fig 4).

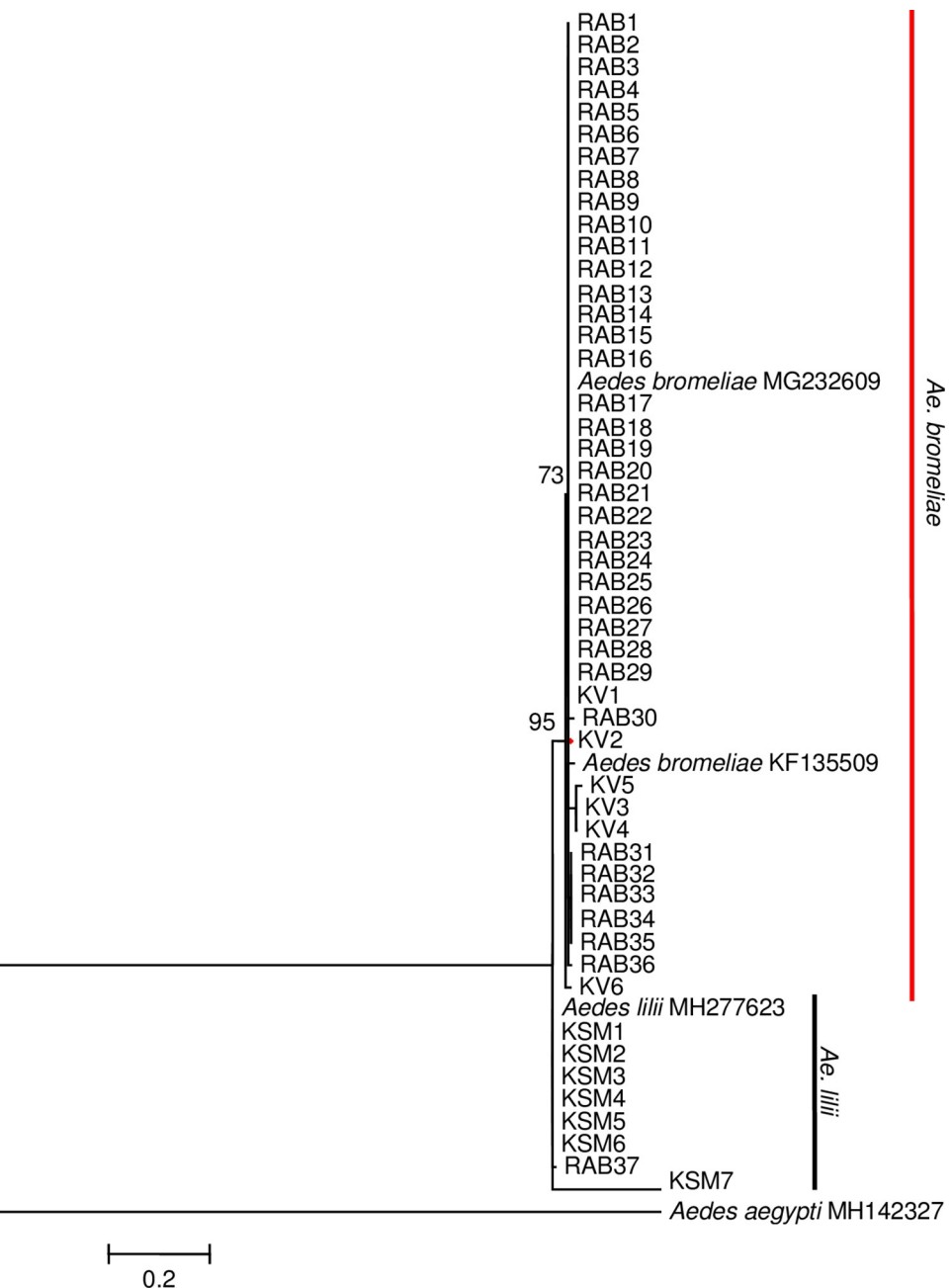

**Fig 2. Maximum-likelihood tree derived from selected parous *Ae. simpsoni* ITS2 sequences using a JC model (216–313 nt).** Bootstrap values are shown above relevant nodes. Sequence of *Ae. aegypti* indicated as outgroup. The scale-bar indicates the number of substitutions per site. Taxon abbreviations represent sampling sites with numbers corresponding to specific sequence samples: RAB, Rabai; KV, Kerio Valley; KSM, Kisumu. Sequences have been submitted to GenBank with accession numbers OK339480–OK339521 (*Aedes bromeliae*), OK339522-OK339529 (*Aedes lilii*).

## Discussion

Here, we provide contemporary estimates of survival and blood feeding patterns for the important arboviral vector *Ae. simpsoni s.l.* collected from two areas with distinct ecosystems. High parity rates and hence survival rates/longevity were evident for this mosquito across the

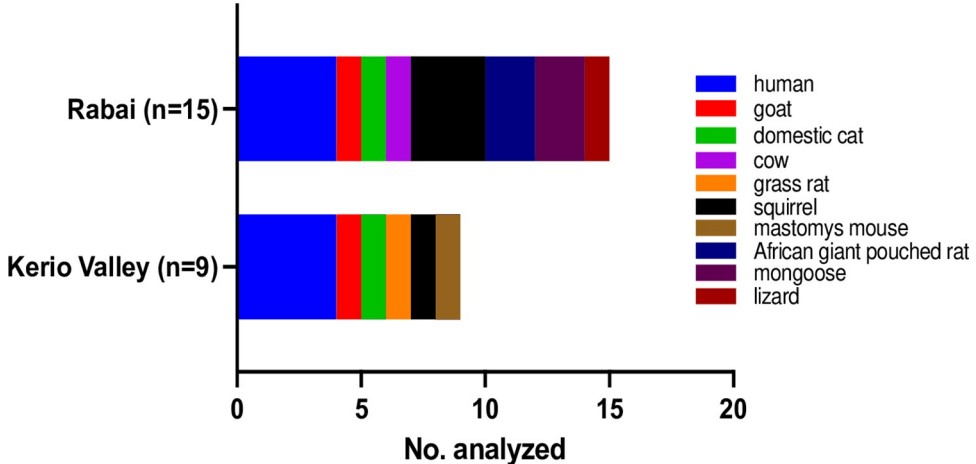

**Fig 3. *Aedes simpsoni* host blood meals.**

three trapping periods. *Aedes simpsoni s.l.* from Kenya was previously reported to be susceptible to YFV [23], and because infectious mosquitoes reflect the age structure of adult female mosquitoes [24], our findings suggest a high vectorial capacity for pathogen transmission by this mosquito species despite moderate human feeding rates.

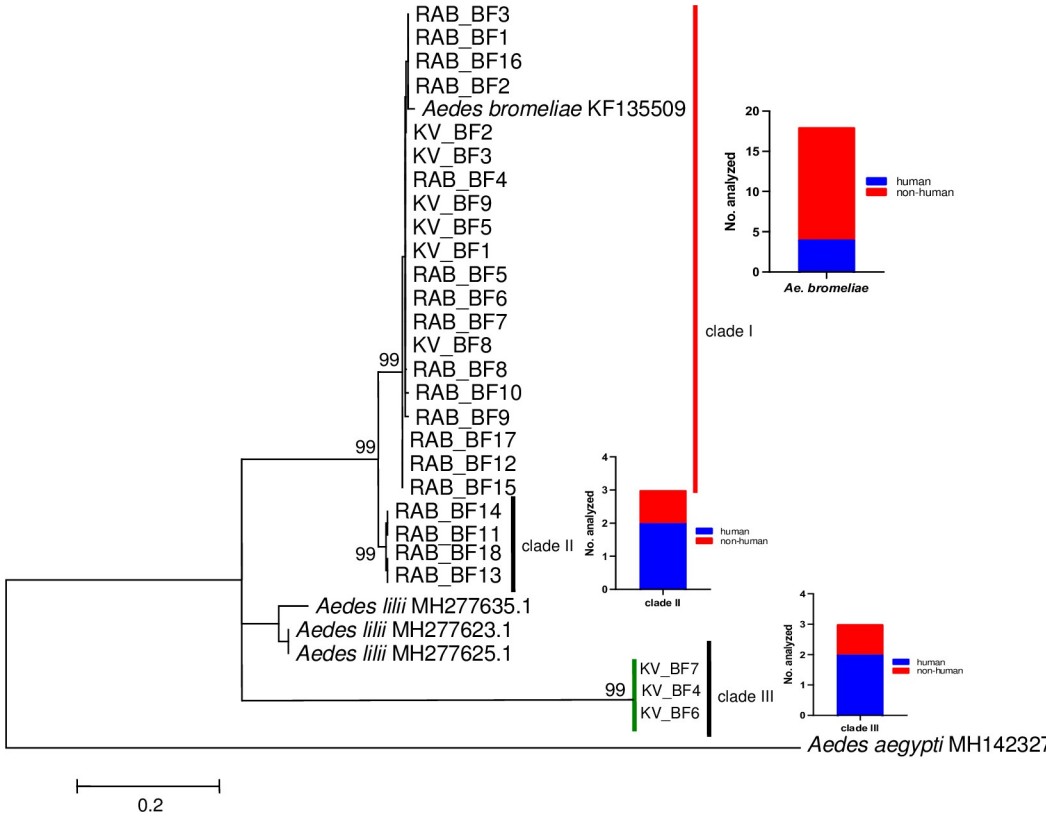

**Fig 4. Maximum-likelihood tree derived from *Aedes simpsoni* blood-fed ITS2 sequences using a JC model (255–313 nt).** Bootstrap values are shown above relevant nodes. Sequence of *Ae. aegypti* indicated as outgroup. The scale-bar indicates the number of substitutions per site. Taxon abbreviations represent sampling sites with numbers corresponding to specific sequence samples: RAB, Rabai; KV, Kerio Valley. Sequences have been submitted to GenBank with accession numbers OK339453–OK339472 (*Aedes bromeliae*), OK339473-OK339479 (*Aedes simpsoni* sl).

*Ae. bromeliae*, which is the only known YFV vector among *Ae. simpsoni* mosquitoes in East Africa [6,25], has equally been suggested as the only human biting sibling species occurring in Kenya [6]. *Ae. bromeliae* appears dominant in the study areas based on our parity data (randomly selected) and blood fed specimens. While our result agrees with data reported in the literature [6,25], this trend might have been affected by its domestic/peri-domestic habits where sampling was conducted. The high survival/longevity found in the current study may contribute to the vectorial capacity of *Ae. bromeliae*. Age is the most sensitive vectorial capacity (a measure of pathogen transmission potential) parameter [8,9]. Higher YFV transmission risk relating to this subspecies is therefore, suggested in Kenya, although validation of its competence is still an important knowledge gap. *Ae. simpsoni* mosquitoes commonly breed in tree holes/plant leaf axils [5]. Its adaptation to breeding in water holding containers around human habitations [5] may contribute to seasonal abundance to enhance human risk as a result of increased vector human contact.

Our sequencing and phylogenetic analyses document for the first time the presence of *Ae. lilii* in Rabai, Kenya, thereby confirming previous speculations on the existence of this subspecies in the area [26]. The presence of *Ae. lilii* in Kenya was further supported by data from Kajulu, Kisumu in western Kenya (Fig 2). The results suggest geographic differences in the distribution of the sibling species in Kenya, warranting further studies to investigate this hypothesis. The very low numbers and absence of blood feeding for *Ae. lilii*, suggest minimal importance of this non-human biter [6,25], in disease transmission to humans in the study areas. Notably, we uncover potentially undescribed species (clades II and III) within the complex based on phylogenetic resolutions which displayed human feeding tendencies (Fig 4). The diversity of species in this complex could be much higher than previously thought which would require additional confirmatory studies. These studies could incorporate additional markers including analysis at the genomic level. Additionally, studies on the distribution and detailed ecology that could facilitate assessment of arboviral disease risk associated with distinct subspecies are warranted. Furthermore, a PCR-based method to distinguish the vector *Ae. bromeliae* from the non-vector *Ae. lilii* based on the ITS region was recently described [25]. Application of this protocol to our samples gave mixed and inconclusive results because several specimens amplified with both specific primer sets targeting these species.

Blood meal data showed feeding on diverse hosts for *Ae. simpsoni* with a corresponding moderate human blood index, a trend largely driven by *Ae. bromeliae*. The overall engorged specimens analyzed was relatively small (n = 24) reflecting the difficulty to trap blood-fed cohorts. This could as well have been affected by our trapping approach possibly biased towards host-seeking females. However, we attempted the newly developed *Aedes* gravid traps to enhance gravid and engorged mosquito cohorts [27] but had no catches. Thus, we suggest that further studies should incorporate resting collections to enhance the number of freshly engorged mosquitoes [28]. While a large sample size could allow for better inference on the trophic habit of these species, nonetheless, our study provides useful baseline data regarding host utilization by this heterogenous species group inferring opportunistic feeding in these arboviral disease foci. Feeding on humans by *Ae. bromeliae* and the potential uncharacterised species, increases the risk of transmission to humans of diverse vector-borne pathogens (e.g., yellow fever, dengue, chikungunya, and Zika viruses) including zoonotic ones circulating in livestock or rodent hosts.

YF outbreak preparedness and the Eliminate Yellow Fever Epidemics (EYE) strategy largely centers on vaccination scale-up, a costly venture that must be guided by justified risk. This focus must expand to include mosquito-based surveillance of critical parameters that define vectorial capacity in specific ecological contexts [8,9]. Combining laboratory experiments and field ecology while incorporating genotype-based analysis of vector populations should

generate integrated data that could be modeled to predict potential YFV spread and re-emergence risk.

We conclude that blood-fed and parous specimens of *Ae. simpsoni s.l.* mosquitoes in the study areas were mainly *Ae. bromeliae* (the primary YFV vector in East Africa), and it appears to be the most dominant species in this complex. High parity rates and hence survival rates/longevity were evident for this mosquito across the three trapping periods, suggesting a high vectorial capacity for pathogen transmission by this mosquito. We report the presence of *Ae. lilii* in Kenya with potentially yet-to-be described species in the complex (clades II and III, Fig 4) displaying human feeding tendencies. We also infer a wide host feeding range on rodents, reptile, domestic livestock besides humans especially for *Ae. bromeliae*, a feeding trend that could likely expose humans to various zoonotic pathogens. Overall, we highlight the importance of precision surveillance data of vector populations through genotype-based analyses for enhanced disease risk prediction and to guide cost-effective interventions (e.g. YF vaccinations).

## Acknowledgments

We thank the chiefs and community members of the study sites for their cooperation and support.

## Author Contributions

**Conceptualization:** Rosemary Sang, David P. Tchouassi.

**Data curation:** Winnie W. Kamau, Gilbert Rotich, David P. Tchouassi.

**Formal analysis:** Winnie W. Kamau, Edwin O. Ogola, David P. Tchouassi.

**Funding acquisition:** Rosemary Sang, Baldwyn Torto, David P. Tchouassi.

**Investigation:** Winnie W. Kamau, Rosemary Sang, Edwin O. Ogola, Gilbert Rotich, Caroline Getugi, Sheila B. Agha, David P. Tchouassi.

**Methodology:** Winnie W. Kamau, David P. Tchouassi.

**Project administration:** Rosemary Sang, Baldwyn Torto, David P. Tchouassi.

**Resources:** Rosemary Sang, Baldwyn Torto, David P. Tchouassi.

**Supervision:** Rosemary Sang, Nelson Menza, David P. Tchouassi.

**Validation:** Rosemary Sang, Nelson Menza, Baldwyn Torto, David P. Tchouassi.

**Visualization:** David P. Tchouassi.

**Writing – original draft:** Winnie W. Kamau, David P. Tchouassi.

**Writing – review & editing:** Winnie W. Kamau, Rosemary Sang, Edwin O. Ogola, Gilbert Rotich, Caroline Getugi, Sheila B. Agha, Nelson Menza, Baldwyn Torto, David P. Tchouassi.

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
