## [Decision Letter · Decision Letter 0]

16 Dec 2021

Dear Dr. Tchouassi,

Thank you very much for submitting your manuscript "Influence of vectoral capacity and genetic structure of Aedes simpsoni on arbovirus transmission risk in East Africa" for consideration at PLOS Neglected Tropical Diseases. As with all papers reviewed by the journal, your manuscript was reviewed by members of the editorial board and by several independent reviewers. In light of the reviews (below this email), we would like to invite the resubmission of a significantly-revised version that takes into account the reviewers' comments. 

Dear Dr. Tchouassi & co-authors,

Many thanks for submitting your manuscript to PLOS NTD. Apologies for the delay in getting this manuscript reviewed. Your manuscript has now been evaluated by three reviewers. The Reviewers agree that the study is of interest and all three provide feedback to improve the manuscript. I would like to ask you to respond to the concerns raised by the Reviewers and revise the manuscript accordingly (major revisions requested). All three reviewers pointed out that the manuscript in its current form, cannot draw firm conclusions regarding vectorial capacity since e.g. the vector competence for YFV of the mosquito species in question is unknown. It would be wonderful if the authors could provide experimental data to assess vector competence. However, if such data cannot be provided in this revision, the title and text of the manuscript should be modify to reflect this. 

Please note that some Reviewers have provided additional comments in a word file. 

Many thanks for your hard work - I look forward to seeing a revised version of this manuscript. 

all the best,

Felix

We cannot make any decision about publication until we have seen the revised manuscript and your response to the reviewers' comments. Your revised manuscript is also likely to be sent to reviewers for further evaluation.

Sincerely,

Felix Hol

Associate Editor

Samuel Scarpino

Deputy Editor

Dear Dr. Tchouassi & co-authors,

Many thanks for submitting your manuscript to PLOS NTD. Apologies for the delay in getting this manuscript reviewed. Your manuscript has now been evaluated by three reviewers. The Reviewers agree that the study is of interest and all three provide feedback to improve the manuscript. I would like to ask you to respond to the concerns raised by the Reviewers and revise the manuscript accordingly (major revisions requested). All three reviewers pointed out that the manuscript in its current form, cannot draw firm conclusions regarding vectorial capacity since e.g. the vector competence for YFV of the mosquito species in question is unknown. It would be wonderful if the authors could provide experimental data to assess vector competence. However, if such data cannot be provided in this revision, the title and text of the manuscript should be modify to reflect this. 

Many thanks for your hard work - I look forward to seeing a revised version of this manuscript. 

all the best,

Felix

Reviewer's Responses to Questions

**Key Review Criteria Required for Acceptance?**

**Methods**

-Are the objectives of the study clearly articulated with a clear testable hypothesis stated?

-Is the study design appropriate to address the stated objectives?

-Is the population clearly described and appropriate for the hypothesis being tested?

-Is the sample size sufficient to ensure adequate power to address the hypothesis being tested?

-Were correct statistical analysis used to support conclusions?

-Are there concerns about ethical or regulatory requirements being met?

Reviewer #1: Objectives are clearly articulated and methods used are appropriate. Sample size is small and imbalanced for certain analysis, and thus prevents some statistical comparisons. However, the data is still informative.

Reviewer #2: The methods are adequate and well described. Sample size and statistics are correct.

However, the study design does not completely test the hypothesis of vector capacity. vector capacity encompasses vector competence and mosquito behavior and the study does not contain a measure of vector competence. Furthermore, there is no mention of earlier evaluation of the Ae bromeliae vector competence.

Reviewer #3: Local climate and environmental/landscape features of studied sites that may promote Ae. simpsoni s.l occurrence and possibly influence mosquito bionomics should be described, such as temperature, rainfalls pattern, humidity, etc, apart from tree holes and plat axile. Since this study has to do with vectorial capacity, which in turns depends on mosquito blood habits and host availability, a general account of human and other domestic animals census data should be described too.

Mosquito survey

Adult host-seeking female mosquitoes were surveyed using CO2-baited BG Sentinel traps at three trapping periods after the rains (lines 124-125). Clarify what do these trapping periods consisted about and why samplings were conducted only after rains? No details have been provided to clarify the procedure used to assign the traps to different sites. 

Lines 126-129, inform of daily variations of temperature, rainfalls, humidity are results actually, so they should be placed in a specific section of the results. 

Line 130 - Traps were set during the day (06:30 – 18:00h). Mosquito sampling as performed from morning to evening (06:30 – 18:00). 

Parity rate, daily survival and longevity estimation

It is not clear whether mosquitoes were dissected after preservation or before preservation at -80ºC.

Statistical analyses

Proper statistical analysis to determine the degree of variability of genetic structure between species should be conducted as well.

**Results**

-Does the analysis presented match the analysis plan?

-Are the results clearly and completely presented?

-Are the figures (Tables, Images) of sufficient quality for clarity?

Reviewer #1: (No Response)

Reviewer #2: The results are clearly and completely presented. However, as mentioned for the methods, the study would greatly benefit from a measure of vector competence (by oral feeding Ae bromeliae with YFV) to meet their objective and main message of identifying Ae bromeliae as a capable vector for YFV.

Reviewer #3: Parity, daily survival and age estimates

I will ask authors to refrain from making statistical analysis to determine the significance of difference between data from different locations when the sampling timings are not similar because it is illogical. For instance, in Rabai the data used to estimate parity rate are from August-September 2019 and February 2020, whereas in KV there was only data from November 2019. There is no point here doing any sort of comparison to determine difference between sites. Do only descriptive analysis that should be restricted to each site. Don’t do any site-to-site comparison because the data were obtained in different contexts, there is no room for spatial comparisons with these data. 

Genetic composition of parous Ae. simpsoni s.l.

This study didn’t characterize mosquito genome, what you did was conducting molecular identification of Ae. simpsoni sibling species by analysing the ITS2 region gene sequences. So, for clarity consider changing the title so it can read 

Species composition of Ae. simpsoni complex.

Line 215, specify a subset of what sample size? How were the sample subsets selected?

Line 218 “…suggesting predominance of this species in both areas”. This is a discussion, not a result. Place it somewhere in the discussion.

Blood meal feeding patterns

Line 231 – indicate the total blood-fed mosquitoes from KV and Rabai that together made up 27 blood-fed specimens.

Discussion

The authors discuss the implications of their findings regarding arbovirus transmission risk at study sites, arguably due to high abundance of Ae. bromeliae, one of the most relevant YF vector in East African region. However, the authors have failed to emphasize while discussing result-by-result that in general the overall size of the mosquito sample was too small, that is, only 539 specimens were collected. This obviously precludes from making any strong conclusion regarding the variables analysed, such as, parity rate and longevity and source of blood meal. No strong inference can be also done on feeding habit when only 27 mosquitoes were analysed. The same applies to survival rate and longevity which also require observation of large number of dissected ovaries. Therefore, the results should be described and discussed very cautiously. 

Lines 290 – 291 “The very low numbers and absence of blood feeding for Ae. lilii found in the current study confirms previous reports for the prevalence of this subspecies in eastern Africa”. There is no clear connection between this paragraph and the one before it. Consider rewriting or deleting.

Lines 294- 296, the authors implies that more sibling species may exist in the Ae. simpsoni group apart from those widely known three species. Please elaborate?

**Conclusions**

-Are the conclusions supported by the data presented?

-Are the limitations of analysis clearly described?

-Do the authors discuss how these data can be helpful to advance our understanding of the topic under study?

-Is public health relevance addressed?

Reviewer #1: (No Response)

Reviewer #2: The study aims to provide information about the vectors for YFV in east africa. The authors collected mosquitoes from 3 geographically distinct areas in Kenya. The mosquito species was identified, their parity rates and the origin of the blood (human or other animals). These information incriminate Ae bromiliae from the Ae. simpsoni complex as a vector for arboviruses in these areas. However, there is a lack of information about the vector competence for Ae. bromaliae for the major arboviruses. From a quick literature search that was not available in the introduction, I found that Ae bromeliae vector competence is partially known although the available studies date back some time ago.

The current study would be more informative about the potential role of Ae bromeliae if they included vector comptence data. Importantly, the authors cannot state that they studied the influence of vector capacity of Ae simpsoni without including a measure of vector competence for YFV.

Reviewer #3: Conclusion

This section reflects the data reported

**Editorial and Data Presentation Modifications?**

Reviewer #1: (No Response)

Reviewer #2: There is a mispelling in the title: "influence of vectorIal capacity".

l. 63-65: I do not understand this sentence.

l. 279-281: I also do not understand this sentence.

l. 235: there is a parenthesis missing.

l. 116: there is another parenthesis missing.

Reviewer #3: (No Response)

**Summary and General Comments**

Reviewer #1: (No Response)

Reviewer #2: (No Response)

Reviewer #3: This study reports survey carried out in some villages in Kenya to understand the vectorial capacity and genetic structure of mosquito members of Aedes simpsoni group, a group comprising highly important, albeit neglected, YF vectors such as Ae. bromeliae. Although the data reported are relevant to broaden our undertesting of YF and associated arbovirus transmission ecological system, I think there are major aspect that must be addressed before this manuscript is considered for publication either in Plos Neglect or elsewehere.

 In my understanding the manuscript title is highly misleading as the study hasn’t actually investigate the influence of vectorial capacity and genetic structure of Ae. simpsoni on arbovirus transmission. What author did was applying molecular analysis tools to investigate species composition of Ae. simpsoni complex. By reading the title I was expecting to find in the main manuscript description information about the amount and distribution of genetic variability between members of the Ae. simpsoni complex and possible link with their ability or not of transmitting arbovirus (YF) at study site. Additionally, the data reported doesn’t allow authors to make any inference about vectorial capacity as they only investigated mosquito blood-feeding habits and survival rate. Therefore, further information needs to be gathered so authors can really estimate vectorial capacity, these information include human-biting density (m), which cannot be inferred by a CO2-baited BG-traps, and human-biting habits (a). The product of these two parameters is the human-biting rate (ma). The human biting habit (a) is a product of human biting frequency (i.e., the length of the gonotrophical cycle) and the human blood index. As such, the manuscript doesn’t show convincingly that Ae. bromeliae has higher vectorial capacity although author have concluded that way. The inference of vectorial capacity has been done based solely on estimates of mosquito survival rate and human blood-feeding habits, lacking other fundamental parameters such as those aforementioned. Furthermore, no investigation of arbovirus infection was conducted on collected samples.

A such, in my perception, this study reports basically the occurrence, species composition, survival rate and blood-feeding habitats of Ae. simpsoni complex members. Therefore, result should be reported and discussed considering only the data produced and possible limitations for making generalization given the type of sampling approach applied and relatively lower number of specimens analysed (N= 539). For instance, only 27 blood-fed mosquitoes were analysed to determine the source of blood-meal. No serious conclusion can be made with such a smaller sample size.

Abstract

Abstract lines 34-36, “We here, address the need for improved understanding of vectorial capacity and genetic influence of Ae. simpsoni mosquitoes. Do the authors mean genetic influence of genetic structure?

Line 37- “Age structure was first estimated by parity in Ae. simpsoni”. This should be rewritten to read Age structure was determined based on the appearance of mosquito ovaries observed by dissection.

Lines 37-38 not clear for what purpose PCR and sequencing was used. What gene(s) or genetic markers were targeted?

Lines 39-41, Inference on vectorial capacity is not that simple, High human blood index doesn’t necessarily mean the species in question has consequently high vectorial capacity. Vectorial capacity is also a direct function of mosquito density relative to humans and human-biting habits (obtained by the product of frequency of mosquito biting and the human blood index also known as index of anthropophagy. This study doesn’t estimate any realistic vectorial capacity, it investigated blood-feeding habits and survival rates of Ae. simpsoni complex sibling species. 

Introduction

This paper is about the influence of vectorial capacity and genetic structure of Aedes simpsoni on arbovirus transmission risk. However, I cannot see the rationale of studying the influence of vectorial capacity and genetic structure of Ae. simpsoni on transmission of arbovirus clearly presented in the introduction, particularly the extent to which arbovirus transmission can be influenced by vectorial capacity and genetic structure. The introduction emphasizes more aspects related to Ae. simpsoni bionomics, and virtually none account about the role of genetic structure on arbovirus transmission has been menioned. Moreover, it is not clearly stated what aspects of genetic structure authors have investigated. In fact, the influence of genetic structure hasn’t been mentioned has one of this study goals. It is clear to me that the goal was to study Ae. simpsoni survival rate and blood-feeding pattern, as it has been stated in lines 92-93.

Aedes simpsoni has been referred interchangeably as Ae. simpsoni s.l. and Ae. simpsoni. Ae. simpsoni has been sometimes referred to as Ae. simpsoni s.s which makes the reading difficulty to follow. Therefore, effort should be done to write the species name clearly so readers can understand the authors points. 

Lines 77 – 78, “…last Kenyan YF outbreak”. I would suggest replacing the words “last Kenyan YF outbreaks” by one of the major YFV outbreak observed in Kenya so far (1992 - 95), as it looks to me that those were the final YF outbreak in Kenya, no outbreak will happen again in the future. This kind of contradicts what has been written in a previous paragraph (lines 72-73), concerning importation of YF cases from RDC and Angola into Kenya, in 2015-2016. 

Line 80- “and this leaves the fate of East Africa unknown”. Suggest replacing by whether the East Africa region may face same fate in the future remains uncertain

Line 85- I think the correct spelling is Ae. lilii

PLOS authors have the option to publish the peer review history of their article (what does this mean?). If published, this will include your full peer review and any attached files.

Reviewer #1: No

Reviewer #2: No

Reviewer #3: No
---

## [Editor Report · Decision Letter 1]

13 Jan 2022

Dear Dr. Tchouassi,

Thank you very much for submitting your manuscript "Survival rate, blood feeding habits and sibling species composition of Aedes simpsoni complex: implications for arbovirus transmission risk in East Africa" for consideration at PLOS Neglected Tropical Diseases. As with all papers reviewed by the journal, your manuscript was reviewed by members of the editorial board and by several independent reviewers. The reviewers appreciated the attention to an important topic. Based on the reviews, we are likely to accept this manuscript for publication, providing that you modify the manuscript according to the review recommendations. 

Dear authors,

Many thanks for revising the manuscript and answering the Reviewer's questions. I have some very minor comments that need to be addressed before acceptance, please find them below:

PNTD-D-21-01450_R1

Feedback on revised MS

Abstract:

lines 42-43 “and consequently, high vectorial capacity for pathogen transmission”: please rephrase ‘consequently’ to ‘potentially’ or similar. As the Reviewers indicated, caution should be taken in this study with regards to conclusions related to vectorial capacity (especially in the abstract). The word ‘consequently’ should therefore be revised. Same for line 72 in Author Summary. 

MM

line 144, missing space between with and trapping. 

Results:

line 279, parenthesis missing after mongoose.

Sincerely,

Felix Hol

Associate Editor

Samuel Scarpino

Deputy Editor

Dear authors,

Many thanks for revising the manuscript and answering the Reviewer's questions. I have some very minor comments that need to be addressed before acceptance, please find them below:

PNTD-D-21-01450_R1

Feedback on revised MS

Abstract:

lines 42-43 “and consequently, high vectorial capacity for pathogen transmission”: please rephrase ‘consequently’ to ‘potentially’ or similar. As the Reviewers indicated, caution should be taken in this study with regards to conclusions related to vectorial capacity (especially in the abstract). The word ‘consequently’ should therefore be revised. Same for line 72 in Author Summary. 

MM

line 144, missing space between with and trapping. 

Results:

line 279, parenthesis missing after mongoose.

Figure Files:

Data Requirements:

Reproducibility:

References

---

## [Editor Report · Decision Letter 2]

14 Jan 2022

Dear Dr. Tchouassi,

We are pleased to inform you that your manuscript 'Survival rate, blood feeding habits and sibling species composition of Aedes simpsoni complex: implications for arbovirus transmission risk in East Africa' has been provisionally accepted for publication in PLOS Neglected Tropical Diseases.

Best regards,

Felix Hol

Associate Editor

Samuel Scarpino

Deputy Editor

Many thanks for taking care of these additional edits. Congrats on a nice manuscript - we're happy to publish it.

Felix

---

## [Editor Report · Acceptance letter]

18 Jan 2022

Dear Dr. Tchouassi,

We are delighted to inform you that your manuscript, "Survival rate, blood feeding habits and sibling species composition of Aedes simpsoni complex: implications for arbovirus transmission risk in East Africa," has been formally accepted for publication in PLOS Neglected Tropical Diseases.

Best regards,

Shaden Kamhawi

co-Editor-in-Chief

Paul Brindley

co-Editor-in-Chief
